# MULTI-SHOT CHARACTER CONSISTENCY FOR TEXT-TO-VIDEO GENERATION

A circus dog's life: (1) playing (2) splashing in a pond, and (3) jumping through hoops of fire

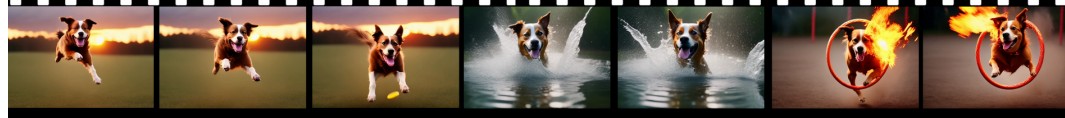

A balloon-person's day: (1) biking to work (2) partying, and (3) hangover on the way back

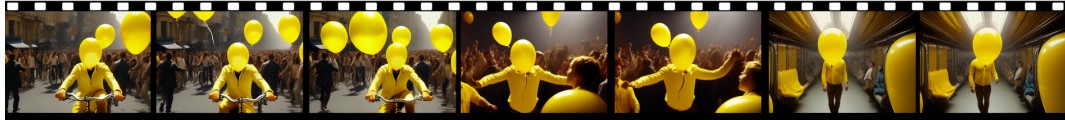

A telekinetic toddler: (1) practicing (2) napping in a cryopod, and (3) controlling nanobots

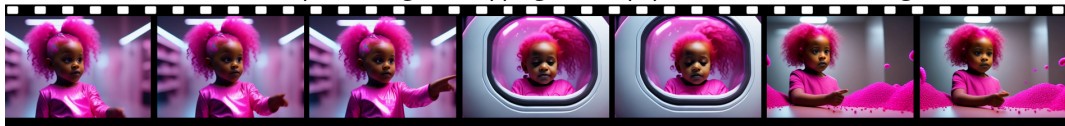

Figure 1: **Consistent Video Storyboarding.** (click-to-view-online) Our method generates video shots from input prompts, ensuring consistent subjects across shots.

## ABSTRACT

Text-to-video models have made significant strides in generating short video clips from textual descriptions. Yet, a significant challenge remains: generating several video shots of the same characters, preserving their identity without hurting video quality, dynamics, and responsiveness to text prompts. We present *Video Storyboarding*, a *training-free* method to enable pretrained text-to-video models to generate multiple shots with consistent characters, by sharing features between them. Our key insight is that self-attention query features (Q) encode both motion and identity. This creates a hard-to-avoid trade-off between preserving character identity and making videos dynamic, when features are shared. To address this issue, we introduce a novel query injection strategy that balances identity preservation and natural motion retention. This approach improves upon naive consistency techniques applied to videos, which often struggle to maintain this delicate equilibrium. Our experiments demonstrate significant improvements in character consistency across scenes while maintaining high-quality motion and text alignment. These results offer insights into critical stages of video generation and the interplay of structure and motion in video diffusion models.

## 1 INTRODUCTION

Generating videos from text prompts is advancing rapidly, but it is still not feasible to create long coherent video sequences. A natural alternative would be to generate multiple short videos that share the same characters. Indeed, cinematic videos typically consist of many shorter shots, making them more engaging. The challenge is that although current text-to-video (T2V) models excel at generating individual clips, they struggle to maintain character consistency across multiple scenes. We wish to generate multiple video shots of the same characters, preserving their identity across

all video scenes, while preserving video quality and dynamics. Such technology would open new opportunities in content creation storytelling and entertainment.

Recently, several methods (Tewel et al., 2024; Fan et al., 2024) have been proposed to generate images with consistent characters across various text prompts. However, achieving consistency in video is inherently more challenging due to a fundamental conflict between maintaining character identity and ensuring dynamic motion. In video, the same features often encode both identity and motion. Thus, when motion occurs, models could interpret it as a change in identity. This problem is unique to video and has not been explored in the context of images. Consequently, existing image-based consistency methods struggle to generalize to video, failing to achieve both character consistency and dynamic motion simultaneously.

We present *Video Storyboarding*, a training-free method for generating multi-shot videos with consistent characters. Our approach uses pre-trained T2V models by sharing features between the video shots. We first demonstrate that the self-attention query (Q) components primarily encode motion information, but they also contain identity features of the generated characters. When features are shared, injecting inconsistent Q components across videos preserves motion but disrupts character identity. Conversely, maintaining similar Q components across generated videos ensure character identity but unifies motion.

To address this, we propose a two-phase approach (Fig. 2): *Q-Preservation* followed by *Q-Flow*. In the Q Preservation phase, we maintain motion structure by replacing our Q values with those from "vanilla" (unconstrained) video generation. Then, the subsequent Q-Flow phase aims to maintain the optical flow of *vanilla* queries rather than their exact values. It employs flow maps derived from vanilla key frames to guide the injection of our identity-preserving Q features. This ensures character identity is maintained by placing consistent features in motion-appropriate locations. Finally, we combine this with a frame selection strategy for extended attention, that promotes visual coherence without freezing motion.

Through extensive ablation studies and comparisons with baseline methods, we show that *Video Storyboarding* significantly improves character and object consistency across scenes while maintaining high-quality motion and adhering to the input text prompt. Our ablation studies provide insights into the critical stages of video generation, the relationship between structure and motion, and the impact of different consistency strategies on the final output.

Our contributions are: (1) *Video Storyboarding*, a novel training-free method that enables character consistency in generating multi-shot video sequences, while maintaining motion adherence to prompts. (2) We reveal the dual role of self-attention query (Q) features in encoding both motion and identity information. (3) We propose a novel two-phase query injection mechanism to balance these aspects, addressing the unique challenges posed by the temporal dimension. (4) We demonstrate significant improvements in character consistency while maintaining motion quality over baselines, tested with two pretrained models.

## 2 RELATED WORK

**Consistent generation** aims to maintain consistent subjects across outputs produced by a generative model. This task has typically been considered under through the lens of text-to-image generation. A common approach is to leverage personalization (Gal et al., 2022; Ruiz et al., 2022) to promote consistency, either through inptainting with a personalized model (Jeong et al., 2023), by iteratively generating multi-character images using personalized LoRA models (Ryu, 2023), or by clustering randomly generated images and training LoRAs for large, semi-consistent clusters (Avrahami et al., 2023). Rather than fine-tuning a model, an encoder (Ye et al., 2023; Wei et al., 2023; Gal et al., 2023) can be used to inject an identity at inference time, but encoders require pre-training on large datasets, and struggle to accurately generalize to arbitrary domains. Similar issues arise when working with models that tune the model on storyboard datasets in order to augment it with additional conditioning on sets of image frames (Feng et al., 2023; Liu et al., 2023). Most recently, works (Tewel et al., 2024; Fan et al., 2024) explored character consistency without personalization, employing feature sharing approaches to generate consistent characters across image batches, without tuning or pre-training.

In our work, we explore bringing the training-free, feature-sharing consistency approach to the realm of video generation, with the goal of maintaining a consistent character across video scenes.

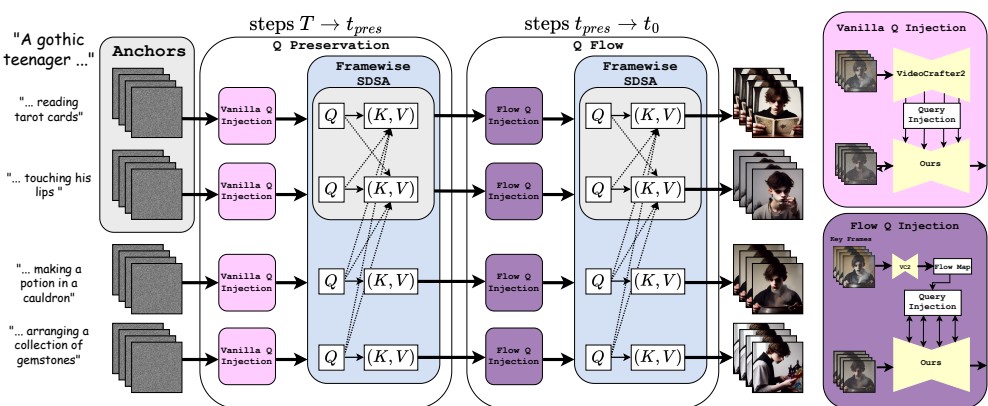

Figure 2: *Video Storyboarding* **Architecture:** Our consistent denoising process has two phases: Q Preservation and Q Flow. We first generate and cache video shots using "vanilla" VideoCrafter2. In Q Preservation ($T \rightarrow t_{pres}$), we use Vanilla Q Injection to maintain motion structure by replacing our Q values with vanilla ones. In Q Flow ($t_{pres} \rightarrow t_0$), we use a flow map from vanilla key frames to guide Q feature injection. This phase maintains character identity by allowing the use of Q features from our consistent denoising process, while the flow map ensures that these identity-preserving features are applied in a way that's consistent with the original motion. Throughout, we employ two complementary techniques: framewise subject-driven self-attention for visual coherence, and refinement feature injection (Section 4.3) to reinforce character consistency across diverse prompts.

**Attention-based consistency.** When using text-to-image models to generate (Wu et al., 2023; Ceylan et al., 2023; Khachatryan et al., 2023) or modify a video (Geyer et al., 2023), an extended self-attention block (Wu et al., 2023) is often employed to share keys and values across different frames, enabling them to draw visual appearances from each other and enhance consistency. Beyond cross-frame consistency, it has been used to inject consistent identities from a source image to video (Xu et al., 2023; Hu et al., 2023; Chang et al., 2023; Tu et al., 2023), maintain appearance in layout editing (Cao et al., 2023; Avrahami et al., 2024), combine appearances (Alaluf et al., 2023), for personalization (Gal et al., 2024; Zeng et al., 2024) and style transfer (Hertz et al., 2023).

In Consistory, Tewel et al. (2024) demonstrate that masked extended-attention can promote subject consistency across batch-generated images. We extend this to video generation, maintaining character consistency between video clips. We also use a masked extended-attention mechanism, but couple it with query-injection components to better maintain motion.

**Text-to-video synthesis.** Following large, diffusion-based (Ho et al., 2020) text-to-image models (Rombach et al., 2021; Ramesh et al., 2022), works sought to replicate their success in video generation. Early text-based video approaches (Ho et al., 2022) leveraged cascaded approaches for time and space super-resolution. Methods commonly leveraged pre-trained text-to-image models' knowledge, expanding them into video models (Wang et al., 2023a; He et al., 2022; Blattmann et al., 2023; Zhou et al., 2022; Wang et al., 2023c; Singer et al., 2023; Luo et al., 2023; Ge et al., 2023; Zhang et al., 2023a; Bar-Tal et al., 2024). Concurrent to such approaches, image-to-video models emerged as powerful alternatives (Gu et al., 2023; Wang et al., 2023b; Zhang et al., 2023b). While not strictly text-conditioned, these can be paired with a text-to-image model to generate an initial frame, which is then animated. Our work builds on existing T2V models (Chen et al., 2024), enabling them to maintain consistent characters across individually generated scenes.

## 3 PRELIMINARIES

### 3.1 NOTATIONS

Our method manipulates spatial self-attention activations in T2V diffusion models. We denote by $\{Q, K, V, O\}$ the respective Query, Key, Value and Output features of a single self-attention layer

(see Appendix A.3 for background). In our method, these features interact across frames, enabling cross-frame attention and consistency. We denote by $Q_v$ the $Q$ features of a layer during a "vanilla", non-consistent, forward pass in a pretrained network, $Q_c$ the query features from our subject-consistent model, and $Q_f$ as the flow-based query features. For brevity, we omitted the frame index $i$

## 3.2 Training-Free Consistent Text-to-Image generation.

Our work extends ConsiStory (Tewel et al., 2024) from image to video generation, addressing the interplay between motion dynamics and identity. ConsiStory operates in three steps. **(1) Subject-Driven localization with extended Self-Attention (SDSA) –** localizes the subject across a set of noisy generated images by aggregating cross-attention maps across layers and timesteps. To ensure subject consistency, SDSA enables each image to attend to patches of the main subject present in *other* image frames. This is done by extending the self-attention mechanism, allowing it to share K, V features of the subject between multiple images. Unfortunately, SDSA alone diminishes *layout* diversity in the generated images. Therefore, **(2) Layout Diversity –** reinforces diversity through two techniques: First, it incorporates Q features from a vanilla, *non-consistent* sampling step. Second, it applies an inference-time dropout to the shared K, V features. Finally, **(3) Refinement Injection –** improves consistency in finer details by injecting the O features between corresponding subject patches.

To reduce computational complexity, and enable reusable subjects, ConsiStory uses "anchor images": Non-anchor images observe features from anchors during generation, but not vice versa.

## 3.3 Flow-based feature injection

Our approach draws inspiration from TokenFlow Geyer et al. (2023), a method for text-guided *video* editing. TokenFlow enforces temporal consistency in the diffusion feature space by propagating features across frames based on inter-frame correspondences. For each feature at a given location in frame $f$, TokenFlow identifies similar features in two nearby keyframes and creates a new feature by blending them according to the frame's relative position. This preserves the overall motion pattern from the input video being edited, while incorporating the text-guided modifications.

## 4 Method

Our goal is to generate multiple video shots that consistently portray the same character across different scenarios described by text prompts. Building on ConsiStory, it addresses the unique challenges inherent in video generation, particularly the preservation of motion dynamics alongside character consistency. Our method comprises three key components: Framewise SDSA, a novel Query injection for motion guidance, and a Deartifacted Refinement Injection. These components work together to generate multiple video shots with consistent character and ensure motion fidelity. The pipeline is illustrated in Fig. 2.

**Vanilla Caching:** Before initiating our main pipeline, we generate a set of video shots using a vanilla pretrained T2V model and cache the query features, $Q_v$, of each denoising step.

### 4.1 Framewise Subject-Driven Self-Attention

Our first step builds on the Subject-Driven Self-Attention (SDSA) mechanism (Sec. 3.2) to incorporate subject features across multiple video shots by extending the self-attention mechanism. We identified two critical challenges when adapting SDSA to video generation: (1) reliably localizing the subject during video denoising, and (2) ensuring motion fluidity is not compromised.

For subject localization, we propose using the estimated clean image $\hat{x}_0$ for mask generation instead of relying on internal network activations, ensuring reliable masks even in early denoising steps. For motion fluidity, we introduce a framewise attention scheme, where frames with matching temporal indices across shots selectively attend each other. This prevents artifacts and frozen motion.

We term this component Framewise-SDSA. Further technical details, including the mask estimation process and the formal definition of Framewise-SDSA, are provided in Appendix A.4..

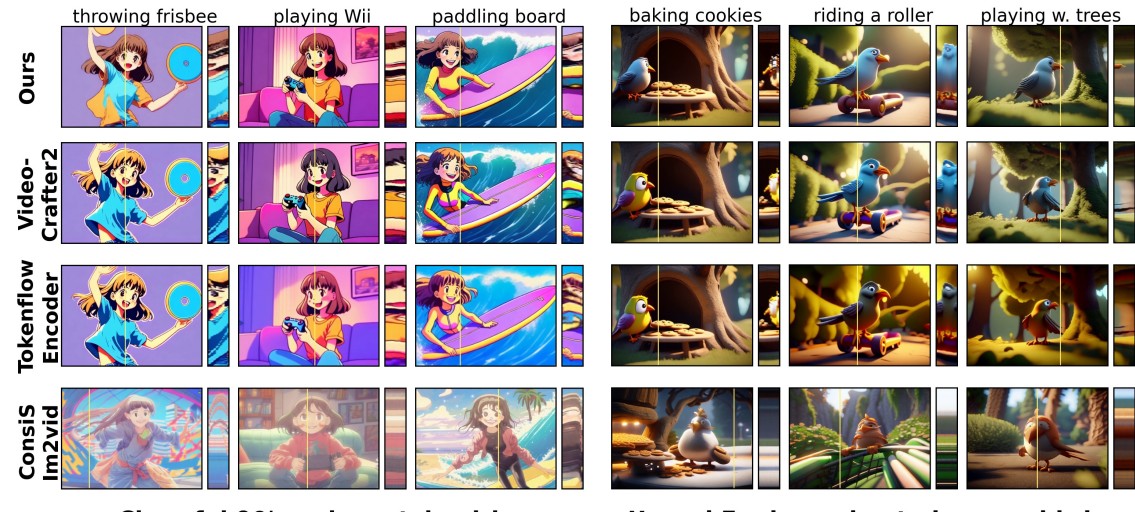

Figure 3: **Qualitative Comparisons.** (click-to-view-online) The first frame of each video shot is displayed along with a spatiotemporal y–t slice to visualize motion. *Ours (top row)* shows improved character consistency across shots while maintaining natural motion. *VideoCrafter2* (row 2) is the vanilla model, showing diverse motion but inconsistent characters. *Tokenflow-Encoder* (row 3) preserves original motion but struggles with character consistency and introduces coloring artifacts. *ConsiS Im2Vid (bottom row)* fails to maintain consistency and exhibits limited motion adherence to prompts. See more examples in Fig. 11.

### 4.2 TWO-PHASE QUERY INJECTION FOR MOTION PRESERVATION

When generating multiple video shots with consistent subjects, we face a fundamental trade-off between subject consistency and motion quality. Our experiments show that while Framewise-SDSA improves subject consistency, it often results in side-effects, leading to excessive synchronization of motion layout across video shots and introduces motion artifacts (Fig. 4(4th row)). These artifacts arise from the model's attempt to simultaneously satisfy both the text prompt and the undesired synchronization across shots.

Prior work in ConsiStory (Sec. 3.2) demonstrated success in maintaining layout diversity for image generation through SDSA dropout and query injection. However, our experiments show that directly extending this approach to video generation produces poor results, with significant visual artifacts and compromised consistency between shots (Fig. 5). This likely occurs because ConsiStory's vanilla-network queries are derived from latents that are influenced by consistency-preserving mechanisms in earlier steps, rather than following an independent denoising trajectory.

Our analysis (Fig. 4) reveals that query features encode both motion patterns and subject identity. Injecting only vanilla query features $(Q_v)$ preserves dynamic motion but results in inconsistent subjects across shots (row 3). Conversely, using only consistency-aware query features $(Q_c)$ ensures subject consistency but produces rigid, unnatural, and synchronized movements (row 4). This observation motivates our two-phase approach that leverages both feature types.

**Phase 1: Motion Structure Establishment.** In early denoising steps ($t \in [T, t_{\text{pres}}]$), we focus on establishing a robust initial motion structure using a process we call Q Preservation. During this phase, we directly inject vanilla query features $(Q_v)$ from pre-generated video shots. This allows us to retain the motion patterns present in the vanilla videos. Without this initial phase, later denoising steps may deviate from the from the original motion patterns, leading to degraded motion quality.

**Phase 2: Flow-based Consistency Integration.** As denoising progresses (beyond $t_{\text{pres}}$), subject consistency becomes increasingly important. To address this, we introduce Q Flow, a technique inspired by TokenFlow Geyer et al. (2023), where flow-based query features $(Q_f)$ are injected to incorporate subject-consistent information while preserving the original motion. Similar to Sec. 3.3, in this phase, we derive a flow map from vanilla-generated keyframes $(Q_v)$, which provides

the motion structure. We then blend subject-consistent query features ($Q_c$) from nearby frames, as dictated by the flow. This blending process produces $Q_f$, that adhere to the original motion patterns while maintaining subject consistency across frames.

By following this approach, we maintain the natural flow of motion established in Phase 1 and progressively integrate subject-consistent features without sacrificing motion quality. The formal definition of our flow-based query injection process is provided in Appendix A.5.

### 4.3 Refinement Feature Injection for Enhanced Consistency

Despite improved motion preservation and subject consistency, fine details in subject appearance can still vary across frames. We address this by adapting the refinement feature injection technique.

However, naively applying refinement feature injection solely to the conditional denoising step, as in ConsiStory, introduces unnatural motion artifacts. This is likely due to the conditional step uses a correspondence map to inject features from different frames, while the unconditional step does not, resulting in inconsistent feature injection. To mitigate this, we extend refinement feature injection to the unconditional denoising step, using the same DIFT correspondence map. We also utilize the entire frame set of each anchor video for refinement injection. This synchronized approach improves overall consistency and reduces motion artifacts. For qualitative results, see Fig. 5.

### 4.4 Implementation details

**Anchor Videos:** Similar to ConsiStory, we use two anchor videos that share all features between themselves. Further implementation details are provided in Appendix A.7

## 5 Experiments

We compare *Video Storyboarding* with strong baselines, starting with a qualitative comparison that shows improved subject-consistency and better motion-alignment. We then conduct an ablation study to examine how self-attention query (Q) tokens affect motion and identity, highlighting the contributions of the components in our method. Finally, quantitative evaluation follows, including a large-scale user study, which demonstrates that users typically favor our results.

### 5.1 Evaluation baselines

We compare our method to three baselines: **(1) VideoCrafter2:** A baseline "vanilla" text-to-video model (Chen et al., 2024), without adaptations. VideoCrafter2 is a public SoTA video model (Huang et al., 2024). **(2) Tokenflow-Encoder:** A combination of TokenFlow Geyer et al. (2023) with IP-Adapter, a Personalization-Based Encoder (Ye et al., 2023). We personalize TokenFlow by conditioning the IP-Adapter on the first frame of one video generated by the vanilla model. For IP-Adapter we use a high-scale hyper-parameter to push the model toward stronger consistency. **(3) ConsiS Im2Vid:** A combination of SoTA *image*-consistency approach (Tewel et al., 2024), with a subsequent Image-to-Video variant of VideoCrafter (Chen et al., 2024). First, we generate a set of consistent *reference* images. Then, we use them as inputs to an Image-to-Video model. We chose VideoCrafter, as it is a public image-to-video model that has an overall quality equivalent to that of the text-to-video VideoCrafter2 model according to the VBench benchmark Huang et al. (2024). **(4) VSTAR:** A method for generating a long video with dynamic evolution (Li et al., 2024b). We directly provide the multiple prompts and sample 16 frames per prompt, then splitting the result into individual shots. **(5) Turbo-V2:** A recent state-of-the-art text-to-video model Li et al. (2024a) that we use to demonstrate our method's adaptability to other architectures.

### 5.2 Qualitative Results

To visually assess both multi-shot consistency and motion quality in videos, we present two elements per video shot: the initial frame for comparing consistency between shots, and a spatiotemporal slice of the space-time volume, termed "y–t slice" Cohen et al. (2024), to visualize motion quality. The selected column for the y–t slice is marked by a yellow line. Typically, we choose the column with

the maximum variance in the vanilla-generated video shot. Occasionally, we manually select the y–t column to highlight specific motion characteristics. For ConsiS Im2Vid, the max-variance column is chosen independently, as it does not directly correspond to the vanilla model.

In Fig. 3 and Fig. 11, we showcase qualitative comparisons between our approach, the vanilla model, and the baselines. Our method demonstrates the ability to alter subject identities consistently across shots, while guiding them towards a unified appearance. This consistency is evident when comparing image frames from different shots. Additionally, an examination of the y–t motion slices reveals that our approach successfully adheres to the motion guided by the vanilla model.

**We encourage readers to visit our online anonymous website at** https:// videostoryboarding.github.io/ for a comprehensive presentation of our results. This website includes playable videos showing our figures in motion, offering a clearer demonstration of our work. For local viewing, extract the attached supplemental zip, then open `index.html`.

The Tokenflow-Encoder baseline preserves the original motion from vanilla models while primarily affecting the color palette and color style of objects and scenes in videos. However, its impact on the identity of the subject is less pronounced than our approach. Additionally, the combination with a high-scaled IP-Adapter often degrades video quality, causing blurring and color artifacts. See the bird example in Fig. 3 (3rd row) and the boy in Fig. 11 (3rd row).

The ConsiS Im2Vid baseline maintains consistency in its *reference* images. However, the subsequent image-to-video model introduces certain limitations. It lacks awareness of the consistency requirement and the capability to maintain it, causing the subject identity to vary between video shots. Although consistency is maintained within each shot, overall consistency with the reference image is compromised, as seen in the bird example in Figure 1 (4th row). Additionally, the image-to-video model fails to account for the action specified in the text prompt. This results in either minimal motion or movement that does not align with the prompt, as the model relies solely on the conditioning image and cannot effectively utilize the textual information. See the limited motion in the y–t slices in Fig. 3 (4th row) and the corresponding videos in the supplemental material.

VSTAR (Fig. 11, Appendix) produces large motion dynamics, but struggles with prompt control, often resulting in entire videos misaligning with text descriptions. As it maintains consistency through continuous video generation, it better suits scene transitions than independent shots.

In the appendix, we present additional capabilities of our method. When applied to Turbo-V2 (Fig. 8), our method enables subject consistency while leveraging Turbo-V2's enhanced motion capabilities. Fig. 9 highlights our ability to handle general subject categories , such as "woman". Fig. 10 demonstrates the ability to render multiple subjects consistently in the same scene.

### 5.2.1 ABLATION STUDY

We conducted an ablation study to investigate the effects of self-attention query (Q) tokens on motion and identity, and to highlight the contributions of our method's components.

Fig. 4 illustrates typical generations for different interventions on Q tokens when combined with the extended self-attention mechanism of Fig. 4.1. When we do not intervene in the Q tokens (Fig. 4, 4th row - "No Q Intervention"), subject identity is well-maintained across video shots, but motion quality significantly degrades. This manifests in: 1) Motion synchronization: movements become synchronized across video shots, *e.g.*, the dog's head turning simultaneously in all shots. 2) Reduced variability in motion style and pose: similar actions are repeated across shots, *e.g.*, the dog's leap, the Muppet's centered swaying, the camera movement becomes static in the skating Muppet shot. 3) Motion artifacts: to reconcile the reduced motion variability with each scene's text prompt, videos tend towards motion-artifacts. For example, the skating Muppet's body appears frozen while its legs are displaced to visually accommodate the "skating" action. In contrast, combining extended attention with injected Q tokens cached from vanilla diffusion-sampled video shots (Fig. 4, 3rd row - "Full Q Preservation") restores motion but largely loses subject identity. For instance, the Muppet's colors revert to those of the vanilla model.

These observations **reveal the dual nature of Q tokens**, which is central to the tokens' role in the generation process. Injecting vanilla Q tokens *restores motion*, showing their influence on movement. Simultaneously, it leads to a *loss of subject identity* (e.g., the Muppet's color change), indicating that Q tokens also carry identity information.

Our approach (Fig. 4, 1st row - "Ours") achieves a balance between both worlds. By intervening in the Q tokens throughout the generation process, we restore most of the original motion, including nuanced details like body and face orientations, postures, and natural movement of specific body parts (dog's ears, Muppet's hands and legs). Even the parallax style of video shots is preserved (right Muppet video shot). Our method's effectiveness stems from our two-step process. First, Q preservation in early denoising steps establishes the motion structure before the identity is fully set. Then, the flow-based Q injection allows the Q values to evolve to better match the novel (more consistent) generation, while enforcing some structural alignment through the flow process.

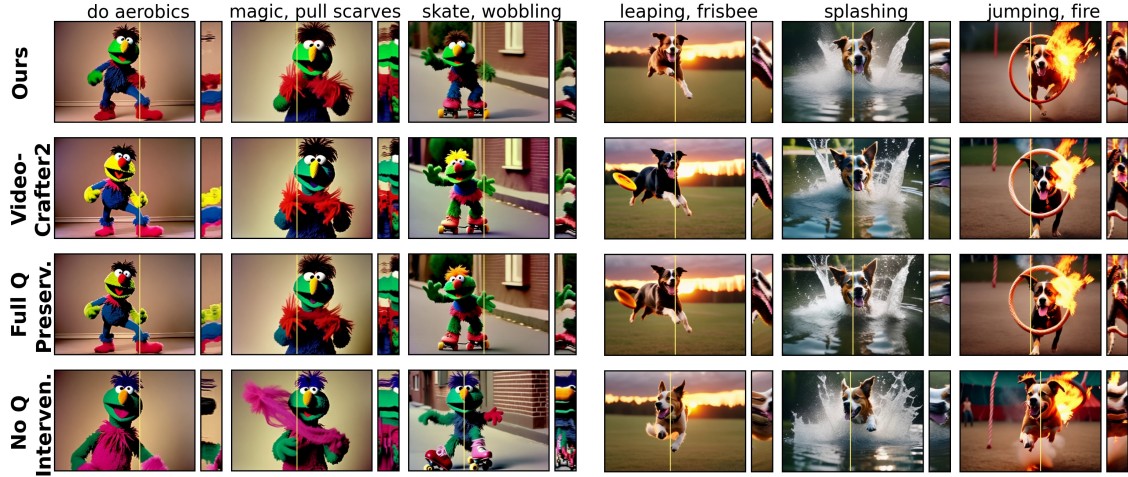

Figure 4: **Ablation Study - comparing Q token intervention strategies.** (click-to-view-online) "Ours" (top row) balances character consistency and natural motion. VideoCrafter2 (second row) shows diverse motion but inconsistent characters. "Full Q Preservation" (third row) directly injects Q tokens from the vanilla model without flow-based processing, preserving original motion but losing character consistency. "No Q Intervention" (bottom row) maintains strong character consistency but suffers from motion degradation and synchronization across shots.

**Adapting ConsiStory for Video Generation.** Next, we demonstrate the challenges of adapting the image-based ConsiStory algorithm Tewel et al. (2024) to video generation. Fig. 5 (3rd row "ConsiS") shows a naive implementation of ConsiStory with subject-driven extended attention coupled across all frames in each video shot, using subject mask dropout and omitting feature injections to the unconditioned diffusion pass. At each step, it also employs queries influenced by the consistency-preserving mechanism of previous steps, rather than queries from an independent vanilla denoising process. This results in impaired identity consistency, strong motion artifacts, and unnatural motion flow of different body parts for both the rabbit and monster examples. Adding feature injection to the unconditional feature denoising (4th row "ConsiS +Uncond") resolves motion artifacts but largely reduces motion magnitude (*e.g.* body postures are mostly frozen), and compromises identity. Next, coupling each frame in a shot with a single frame in an anchor video and avoiding SDSA dropout (5th row "Q ConsiS") allows for subtle natural motion, although it remains partially synchronized. It also improves identity preservation to some degree. Unlike ConsiStory, SDSA dropout in videos hurts identity without significantly improving motion. Finally, our method (1st row - "Ours") employs a novel Q intervention mechanism. It achieves richer motion with better identity and adherence to the original motion of the vanilla model.

## 5.3 QUANTITATIVE EVALUATION

We conducted a quantitative analysis using automated metrics and a user study, based on a benchmark dataset that we created to assess set-consistency in video generation.

**Benchmark Dataset:** We constructed a benchmark dataset of 30 video sets, each containing 5 video-shots with shared subjects but varying prompts. See further details in Appendix A.6.

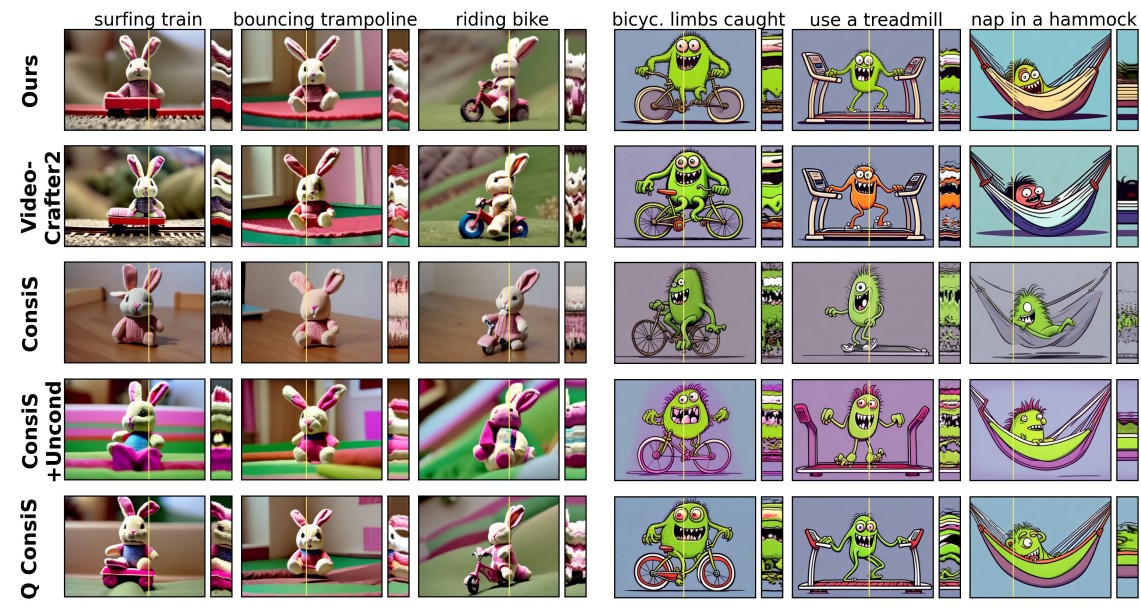

**Camcorder, patchwork stuffed rabbit toy**   **2D cartoon animation, lovable monster**

Figure 5: **Ablation Study on ConsiStory Components for Video Generation.** (click-to-view-online) *"Ours" (top row)* demonstrates improved motion richness and identity preservation. *VideoCrafter2 (second row)* shows diverse motion but inconsistent characters. *"ConsiS" (third row)*, a naive ConsiStory implementation, shows impaired identity and motion artifacts. *"ConsiS +Uncond" (fourth row)* adds feature injection to unconditional denoising, resolving motion artifacts but reducing motion magnitude and compromising identity. *"Q ConsiS" (fifth row)* couples each frame with a single frame in an anchor video, allowing some natural motion, although partially synchronized, with improved identity. Our method achieves the best balance of motion quality and identity.

**Evaluation Protocol:** To avoid overfitting, we conducted all development and parameter tuning on a separate collection of 16 distinct subject-prompt sets. The test set was used exclusively for final evaluations, without any component development or hyperparameter tuning.

**Evaluation Metrics:** Our evaluation approach builds on previous work in image consistency and personalization Tewel et al. (2024); Gal et al. (2022); Ruiz et al. (2022), focusing on multi-shot set-consistency and motion dynamics. For **set-consistency**, we measure average pairwise DINO feature similarity Caron et al. (2021); Huang et al. (2024) across all frames in a set, excluding pairs within the same video shot. We isolate the subject by masking out the background Fu et al. (2023) before extracting each frame's features, using ClipSEG Lüddecke & Ecker (2021) with a dynamic threshold determined by "Otsu's method" Otsu (1979). For **motion dynamics**, we evaluate all 150 generated videos using VBench's "Dynamic Degree" metric Huang et al. (2024), which classifies the significance of video motion by measuring RAFT-based optical flow intensity. We focused on motion dynamics over text prompt alignment due to two challenges: actions are often visible even in videos with minimal motion, making it difficult for temporal CLIP-like models Wang et al. (2024) to distinguish between our method and baselines; also, sharing seeds across baselines lead to similar visual structures, with main differences in motion quality. We include text-similarity metrics in Table 1 (Appendix), measuring temporal CLIP similarity between each video shot and its prompt.

**Results:** Fig. 6 show our approach enhances multi-shot set consistency, while sacrificing motion magnitude compared to vanilla VideoCrafter2. Tokenflow-Encoder baseline shows consistency improvement and slight motion decrease. ConsiS-Im2Vid baseline's performance aligns with qualitative analysis, showing low motion scores. A comparison of all baselines, including VSTAR and Turbo-V2, is presented in Table 1 (Appendix). VSTAR struggles with prompt control (19.8 vs 27.7 for ours), while achieving the highest consistency and motion dynamics. When combined with Turbo-V2, our method improves multi-shot consistency while maintaining high motion quality: The dynamic degree improves threefold, from 20 to 62, while keeping the same level of text alignment.

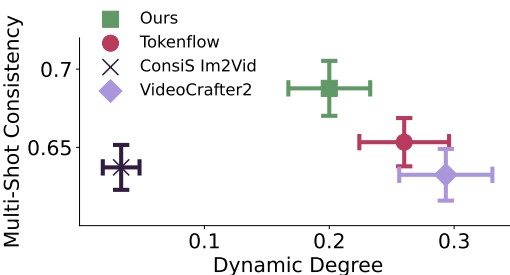

Figure 6: **Quantitative Evaluation of Set Consistency and Motion Dynamics:** Our approach achieves highest set consistency score while maintaining competitive motion dynamics. Error bars indicate standard error of the mean.

These quantitative results offer insights into trade-offs between our approach and baselines, but cannot fully capture user-perceived quality or alignment of generated motions with text prompts. Therefore, we conducted a comprehensive user preference study using two and three-alternative forced-choice format, focusing on two key aspects: set-consistency and text-motion alignment. For set-consistency, users selected the better set from two sets of 5 videos each depicting the subject. For text-motion alignment, users chose the video best matching the action described in the prompt from a pair of videos. To distinguish between degraded motions and those largely unchanged, users could also indicate if motion quality was equivalent in both videos. We used the same test benchmark as the automated metric study, collecting 5 repetitions per question for set-consistency and 3 repetitions for text-motion alignment, totaling 1800 responses.

The user-study results in Fig. 7, reveal that *Video Storyboarding* outperforms the baselines in set consistency. For motion quality, 55% of users rated the generated motions as similar or superior to those of the vanilla model. The ConsiS-Img2Vid baseline's motion quality was consistent with our earlier findings, showing lower motion quality. However, it achieved the highest set consistency among the baselines, winning in 34% of the generated sets compared to our approach.

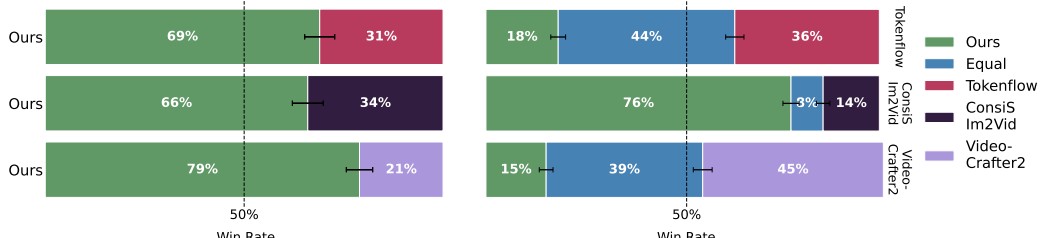

Figure 7: **User Study:** *(left)* We measure user preferences for set consistency and *(right)* how well the generated motion matches the text prompt . Our approach achieves the superior set consistency score while maintaining competitive text-motion alignment. Notably, 55% of our generated motions were judged to be of similar or better quality compared to the vanilla model. Error bars are S.E.M.

## 6 CONCLUSION AND LIMITATIONS

In this work, we introduced *Video Storyboarding*, a novel training-free approach for generating multi-shot video sequences with consistent characters while preserving motion quality. Overall, our method provides a significant step forward in generating coherent multi-shot video sequences, offering a practical solution to the challenge of maintaining character consistency without sacrificing motion quality.

**Limitations:** Our approach was developed for current open text-to-video models which only generate very short videos. It is not known how it will operate with many-second-long videos. Also, balancing identity preservation and motion quality is still not perfect. We find that Q injection may be too strong and still hurt identity. To manage this, motion preservation can be compromised by partially dropping out Q injection (See Section A.2).

REPRODUCIBILITY STATEMENT

Appendix A.7 outlines the details of our implementation, including our technical solution for fitting large batches of video shots within the available GPU memory.

Appendix A.8 provides the exact instructions given to users in the study, along with examples.

The supplemental zip files contain a "prompts" folder, which includes two files: `benchmark_prompts.yaml`, featuring prompts used for experiments with automatic metrics and user studies, and `qualitative_prompts.yaml`, containing prompts for qualitative comparisons and the ablation study.

Appendix A.4 and A.5 describe the mathematical formulation of the framewise-SDSA and Q flow injection components.

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

# A APPENDIX

## A.1 ADDITIONAL RESULTS

Fig. 8 illustrates the adaptability of our method when applied to the state-of-the-art T2V-Turbo-V2 model (Li et al., 2024a). The results show enhanced motion quality while maintaining subject consistency, demonstrating that our approach can effectively improve even the most recent video generation models.

Fig. 9 demonstrates *Video Storyboarding*'s ability to handle general subject categories. The figure shows examples of successfully generating consistent videos for broad subject types like "woman" and "rabbit", indicating the model can work effectively with superclass-level prompts rather than just specific instances. Here we kept the scene style and actions the same as in our other qualitative results, and just changed the subject to a avoid detailed description

Fig. 10 showcases *Video Storyboarding*'s capability to handle multiple subjects. By incorporating two subjects in the prompt of the zero-shot mask, our approach can consistently render multiple characters in the same scene, as demonstrated by examples with girl-owl and boy-teddy bear pairs.

Fig. 11, provides additional qualitative comparisions to Fig. 3, and also includes qualitative comparison with VSTAR baseline (Li et al., 2024b).

In Table 1 we present a comprehensive quantitative comparison across different models using three key metrics. Our method, when combined with both VideoCrafter2 and Turbo-V2, shows improved Multi-Shot Consistency scores (68.8 and 67.3 respectively) compared to their baseline versions (63.2 and 63.3), while maintaining comparable Text Similarity and Dynamic Degree measurements. This indicates that our approach successfully enhances subject consistency without significantly compromising other important aspects of video generation. In the reported metrics, we also include a "Subject-Consistency" metric, introduced by VBench (Huang et al., 2024). This metric measures the similarity between frames within the same video shot using DINO (see Table 1 in the Appendix).

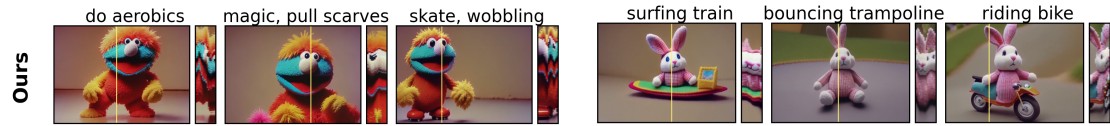

Figure 8: **T2V-Turbo-V2:** (click-to-view-online) *Video Storyboarding* can be applied to T2V-Turbo-V2 Li et al. (2024a), a recent state-of-the-art video model, that exhibits significantly better motion.

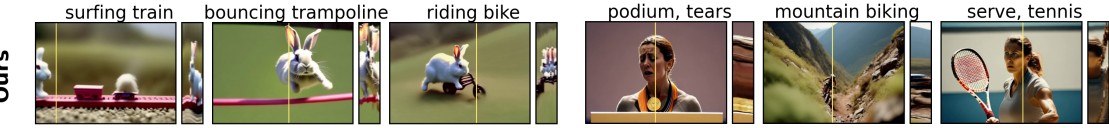

Figure 9: **General subjects:** (click-to-view-online) *Video Storyboarding* can successfully generate consistent subjects when given with general (superclass) subject prompts like woman, or rabbit

## A.2 Q DROPOUT

When Q injection is too strong, it can compromise identity preservation. To address this, we introduce Q dropout, which reduces the strength of Q injection. Unlike SDSA dropout, which hurts identity when trying to improve the image structure, Q dropout sacrifices some visual structural (motion) to enhance identity preservation. This Identity-Motion Trade-off is illustrated in Fig. 12, where increasing Q dropout improves identity consistency but reduces motion richness.

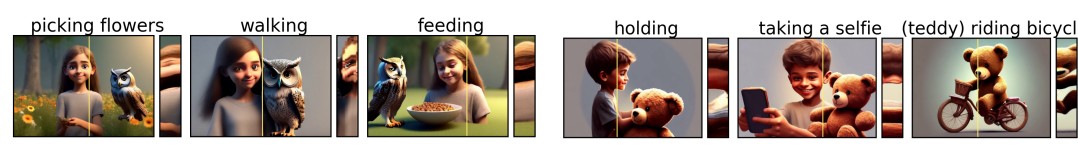

**3D, girl, owl**  **Digital painting, boy, teddy**

Figure 10: **Multi Subject:** (click-to-view-online) By prompting the zero-shot mask with two subjects, our *Video Storyboarding* can render two consistent subjects in a scene.

|  | MULTI-SHOT CONSISTENCY | TEXT SIMILARITY | DYNAMIC DEGREE | SUBJECT CONSISTENCY |
|---|---|---|---|---|
| CONSIS IM2VID | $63.7 \pm 1.4$ | $27.3 \pm 0.5$ | $3.3 \pm 1.5$ | $99.1 \pm 0.1$ |
| VSTAR | $83.9 \pm 1.6$ | $19.8 \pm 0.4$ | $90.7 \pm 2.4$ | $92.6 \pm 0.3$ |
| TOKENFLOW | $65.3 \pm 1.5$ | $27.9 \pm 0.4$ | $26.0 \pm 3.6$ | $97.7 \pm 0.2$ |
| VIDEOCRAFTER2 | $63.2 \pm 1.7$ | $28.7 \pm 0.4$ | $29.3 \pm 3.7$ | $97.3 \pm 0.2$ |
| OURS + VIDEOCRAFTER2 | $68.8 \pm 1.8$ | $27.7 \pm 0.4$ | $20.0 \pm 3.3$ | $97.7 \pm 0.2$ |
| TURBO-V2 | $63.3 \pm 1.7$ | $28.6 \pm 0.4$ | $63.3 \pm 3.9$ | $96.2 \pm 0.2$ |
| OURS + TURBO-V2 | $67.3 \pm 2.1$ | $27.4 \pm 0.4$ | $62.0 \pm 4.0$ | $96.8 \pm 0.2$ |

Table 1: **Quantitative Evaluation Metrics.** Comparison of different models across three metrics: Multi-Shot Consistency, Text Similarity, and Dynamic Degree. Values are reported as mean $\pm$ standard error of the mean (S.E.M).

### A.3 SELF-ATTENTION IN T2V MODELS

Our method manipulates the activations of the spatial self-attention in T2V diffusion models. We start by outlining its mechanism and introducing key notations.

Recent T2V diffusion models are based on a latent video diffusion model (LVDM) architecture where a U-Net denoiser is trained to estimate the noise in the noisy latent codes input. The denoising U-Net is a 3D U-Net architecture consisting of a stack spatio-temporal blocks comprised of convolutional layers, spatial transformers (ST), and temporal transformers (TT). The ST operate independently on each video frame, without awareness of the temporal structure, while the TT operate independently on each temporal patch, without awareness of the spatial structure. In this work, we focus on manipulating the self-attention mechanism of the spatial transformer layers.

Let $x_i \in \mathbb{R}^{P \times d}$ represent the input features for frame $i$, where $P$ is the number of patches and $d$ is the feature dimension. In self-attention, each patch generates three key components: $Q_i$ (*Query features*) to search for relevant information from other patches, $K_i$ (*Key features*) to match against queries, and $V_i$ (*Value features*) containing information to aggregate. The attention map is computed as $A_i = softmax(Q_i K_i^\top / \sqrt{d_k})$, and is used to combine $V_i$ features to produce $O_i$ the final *Output features*: $O_i = W_O \cdot (A_i \cdot V_i)$, where $W_O$ is a linear projection matrix. Our method intervenes in this self-attention mechanism by allowing video-frames in a generated *batch of videos* to attend to each other and be influenced by each other's activations.

### A.4 FRAMEWISE SUBJECT-DRIVEN SELF-ATTENTION - IMPLEMENTATION DETAILS

This section provides a detailed explanation of our proposed Framewise-SDSA mechanism.

**Improved Subject Localization.**  In video generation, subject localization becomes particularly challenging during early denoising steps, where the noise is most prominent. aggregation method proposed in ConsiStory (Sec. 3.2) proved insufficient in this context, particularly during the earliest denoising steps, leading unreliable masks both in terms of accuracy and false positive localization.

To address this, we propose using the estimated clean image $\hat{x}_0$ for subject localization instead of relying on internal network activations. At each denoising step $t$, we estimate $\hat{x}_0$ from the noisy latent

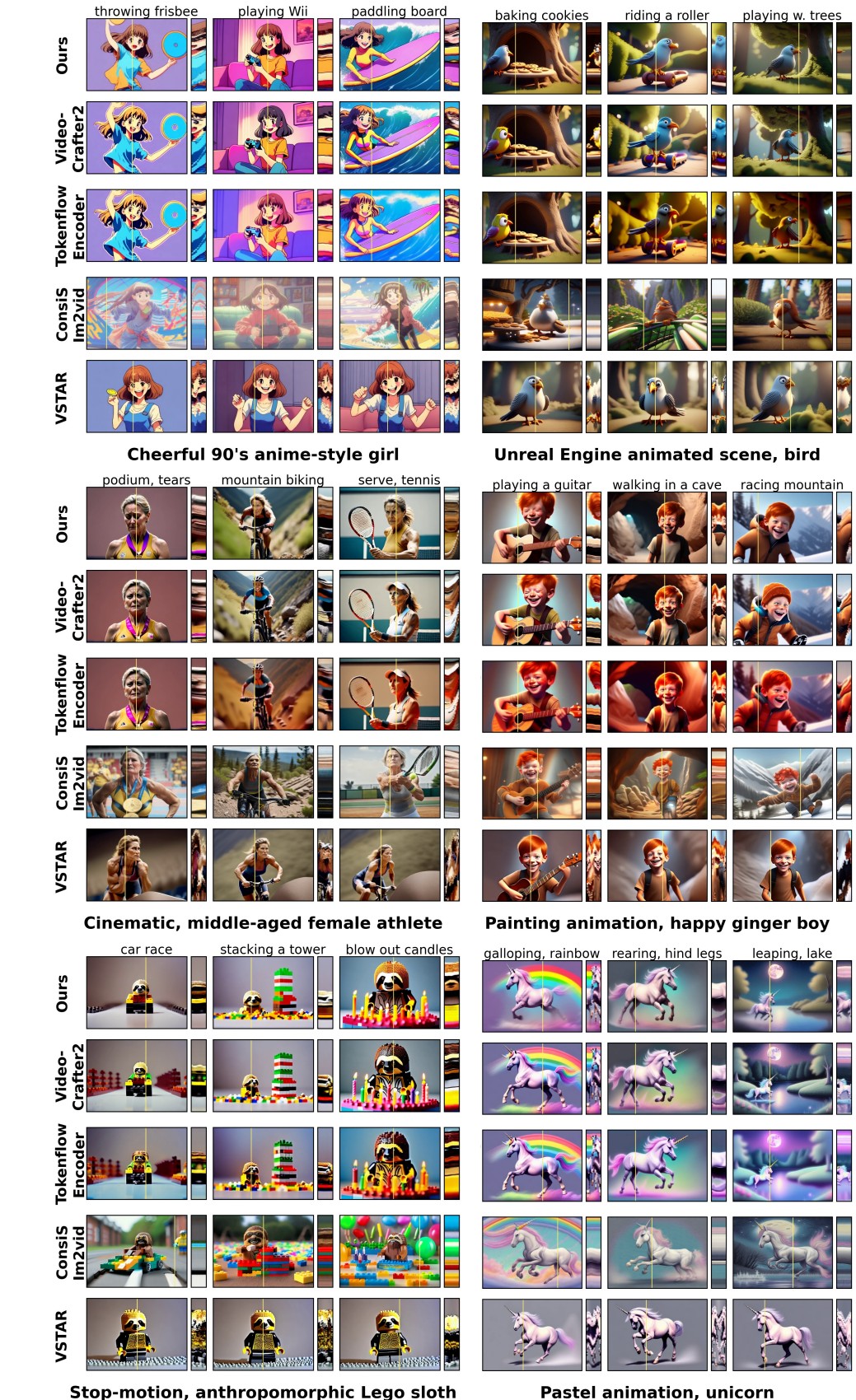

Figure 11: **Additional Qualitative Comparisons , including VSTAR :** (click-to-view-online) Our method generates consistent subjects while preserving diverse and natural motions across scenarios.

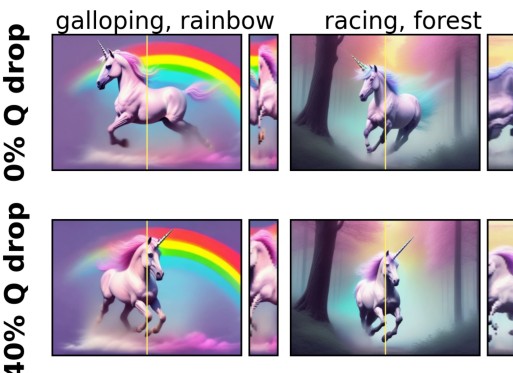

Figure 12: **Q dropout:** Q injection may hurt identity. Q dropout may trade-off identity for motion. At 0% the unicorn gallops at both directions. At 40%, only to the right.

$x$ using: $\hat{x}_0 = \left(x - \sqrt{1-\alpha_t} \cdot e_t\right)/\sqrt{\alpha_t}$, where $e_t$ is the estimated noise, and $\alpha_t$ is the schedule parameter Song et al. (2020). We then apply a zero-shot segmentation approach (Lüddecke & Ecker, 2021) to localize the subject in the estimated image, followed by Otsu's method (Otsu, 1979) to dynamically threshold the mask. This approach produces reliable subject masks from the earliest denoising steps and throughout the generation process.

**Maintaining Motion Fluidity.** Our experiments revealed that a direct application of SDSA – attending to all frames across all videos simultaneously – can lead to visual artifacts and frozen motion. We discovered that limiting attention to a single corresponding frame in other shots is most effective, as attending to two or more frames negatively impacts motion fluidity and introduces visual artifacts. Specifically, we propose a framewise attention scheme. Instead of attending to all frames across all video shots, frames with matching temporal indices across shots attend only to each other. This prevents visual artifacts and frozen motion, which occur when attending to multiple frames simultaneously and strikes a balance between subject consistency and natural motion.

**Formal Definition of Framewise-SDSA.** Let $K_{if}, Q_{if}, V_{if}, M_{if}$ be the keys, queries, values and subject-mask for frame $f$ in video shot $i$. The framewise extended self-attention $A_{if}^+$ is defined by:

$$K_f^+ = [K_{1,f} \oplus K_{2,f} \oplus \cdots \oplus K_{N,f}]$$

$$V_f^+ = [V_{1,f} \oplus V_{2,f} \oplus \cdots \oplus V_{N.f}]$$

$$M_{i,f}^+ = [M_{1,f} \oplus \cdots \oplus M_{i-1,f} \oplus \mathbb{1} \oplus M_{i+1,f} \cdots \oplus M_{N,f}]$$

$$A_{i,f}^+ = softmax\left(Q_i K_f^+ / \sqrt{d_k} + \log M_{i,f}^+\right)$$

$$h_{i,f} = A_{i,f}^+ \cdot V_f^+ \tag{1}$$

where $\oplus$ indicates matrix concatenation. We use standard attention masking, which null-out softmax's logits by assigning their scores to $-\infty$ according to the mask. Note that in this step, the Query tokens remain unaltered, and that the concatenated mask $M_{i,f}^+$ is set to be an array of 1's for patch indices that belong to the $i^{th}$ image itself.

## A.5 FLOW-BASED Q COMPONENTS INJECTION - FORMAL DEFINITION

Let $q_{fxy} \in \mathbb{R}^F$ represent a Q feature from an originally generated video at location $(x, y)$ in frame $f$. We denote by $f_A$ and $f_B$ the indices of the two nearest keyframes, where $f_A \leq f \leq f_B$. The locations of the most similar Q features in frames $f_A$ and $f_B$, denoted by $(x_A, y_A)$ and $(x_B, y_B)$ respectively, are defined as:

$$(x_A, y_A) = \underset{x_0, y_0}{\operatorname{argmax}} \ \mathcal{S}_{\cos}(q_{fxy}, q_{f_A x_0 y_0}) \tag{2}$$

$$(x_B, y_B) = \underset{x_0, y_0}{\operatorname{argmax}} \ \mathcal{S}_{\cos}(q_{fxy}, q_{f_B x_0 y_0}) \tag{3}$$

where $\mathcal{S}_{\cos}(a, b)$ represents the cosine similarity between $a$ and $b$.

We then modify the generated Q feature, denoted by $\hat{q}_{fxy}$, as follows:

$$\hat{q}_{fxy} = w\hat{q}_{f_A x_A y_A} + (1 - w)\hat{q}_{f_B x_B y_B} \tag{4}$$

where $w = \operatorname{sigmoid}\left(\frac{f_B - f}{f_B - f_A}\right)$. This ensures that $\hat{q}$ maintains the feature flow of the originally generated video, without injecting the actual features from it.

## A.6 BENCHMARK DATASET CONSTRUCTION:

We created a benchmark dataset comprising 30 video sets, each containing 5 video-shots depicting a shared subject under different prompts. The evaluation prompts were crafted using the Claude Sonnet 3.5 AI-Agent, following this protocol: each prompt consisted of three parts: (1) a subject description, *e.g.*, *"A girl"* (2) a setting description, *e.g.*, *"paddling out on her surfboard"*, and (3) a style descriptor encompassing both image and motion styles, *e.g.*, *"Anime cartoon animation"* or *"Shaky camcoder footage"*. We instructed the AI-agent to choose actions that are visually striking and could be captured in a split second. Within each set, prompts shared the same subject and style but varied in settings. To ensure a challenging and representative test set, we selected a subset of 5 prompts per subject, prioritizing those that produced videos with significant motion and subject variability when processed by the vanilla model. Importantly, to ensure fairness, this selection process relied solely on the vanilla model's generations.

## A.7 IMPLEMENTATION DETAILS

**Anchor Videos:** Similar to ConsiStory, we utilize two anchor videos that share all features between themselves. Other videos in the batch only observe features derived from these anchors.

**Scalable Video Batch Processing with Sub-batch Attention:** To fit large batches of video generation within available GPU memory, we process the self and cross-attention computations in smaller sub-batches. This approach uses an internal loop, and subsequently concatenates results into a single tensor. The operation remains transparent to the network, enabling the generation of larger batches of video shots.

**Reproducible denoising.** Our pipeline involves three denoising iterations: caching vanilla queries, applying Q injection and Framewise SDSA, and adding refinement feature injection. To ensure consistency across these stages, we maintain identical random generators for both initial noisy latents and the denoising process. This approach guarantees that each part builds upon the previous one, preserving the reliability of our reproducible denoising pipeline.

**Temporal Parameters:** For Q preservation, we set $t_{pres}$ to 750. Framewise-SDSA is applied for $t \in [550, 950]$. Our refinement feature injection step is employed during $t \in [590, 950]$.

**Feature Injection:** We apply our refinement feature injection step to the $32 \times 20$ self-attention layers. Other layers either produced visual artifacts or did not significantly affect identity.

**T2V-Turbo-V2:** For T2V-Turbo-V2 we adapt our Framewise-SDSA by allowing each frame to attend to both its temporally matching frames across shots and the middle frame of each shot. Other hyper-parameters were kept the same.

## A.8 USER STUDY PROTOCOL

The following screenshots illustrate the experimental framework used in our user study:

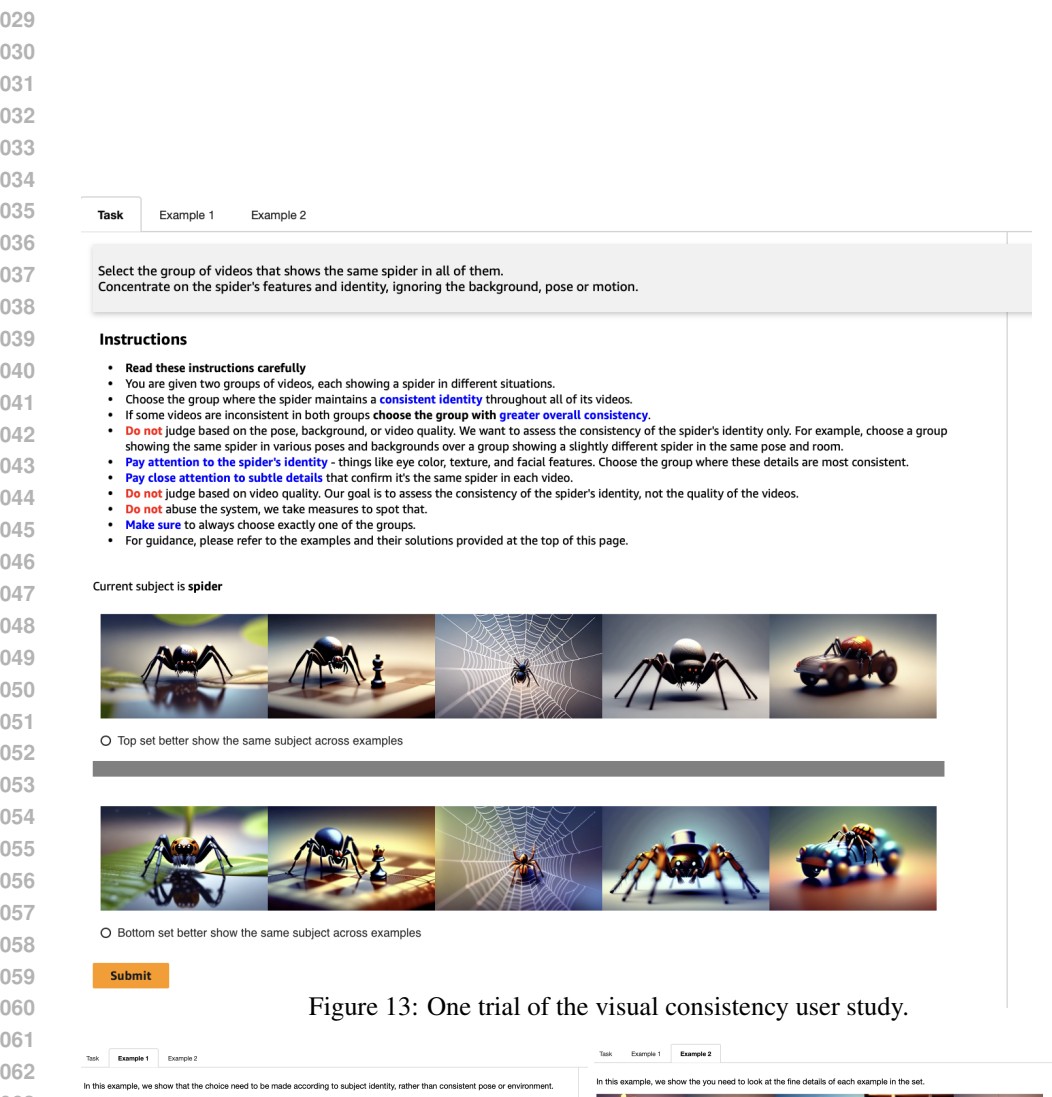

Figure 13: One trial of the visual consistency user study.

Figure 14: Examples provided in the user study for visual set consistency.

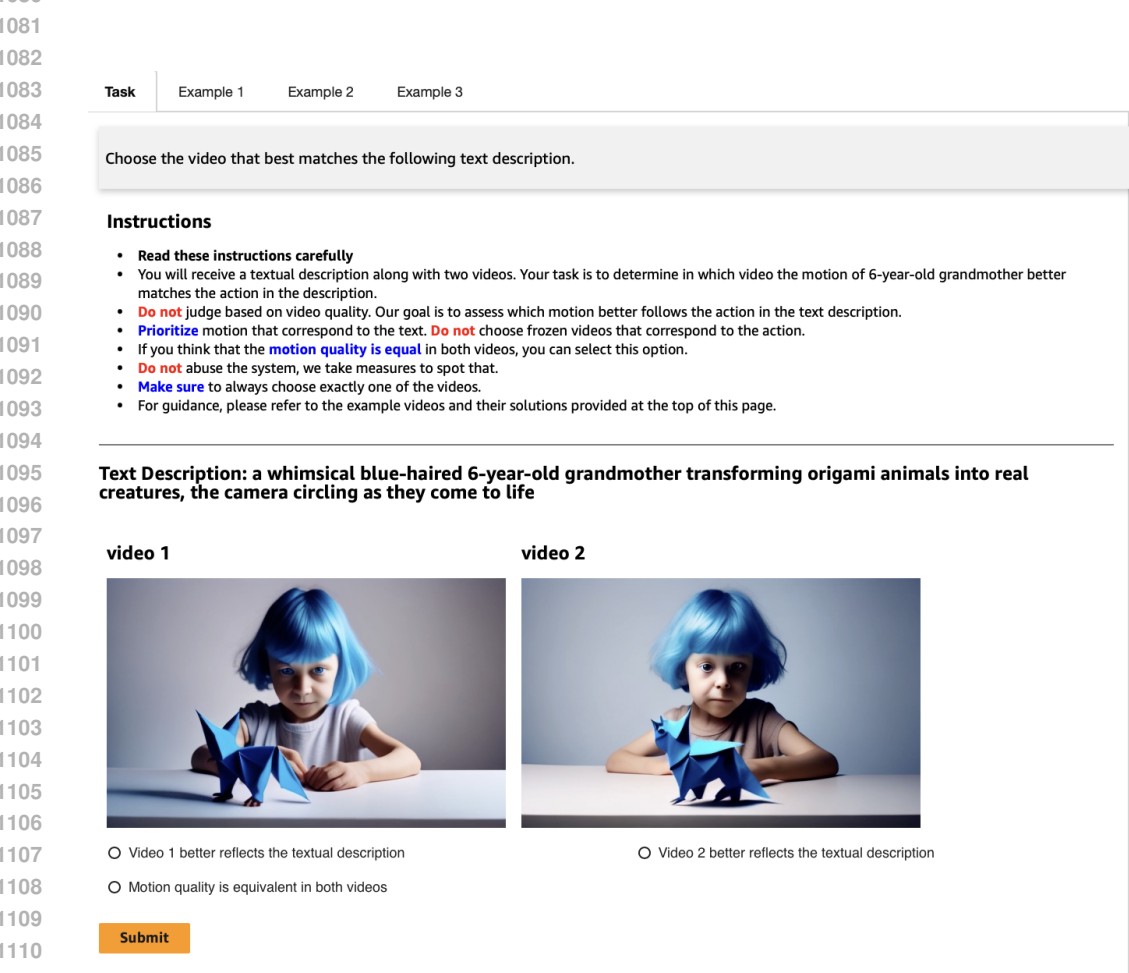

Figure 15: One trial of the text-motion alignment user study.

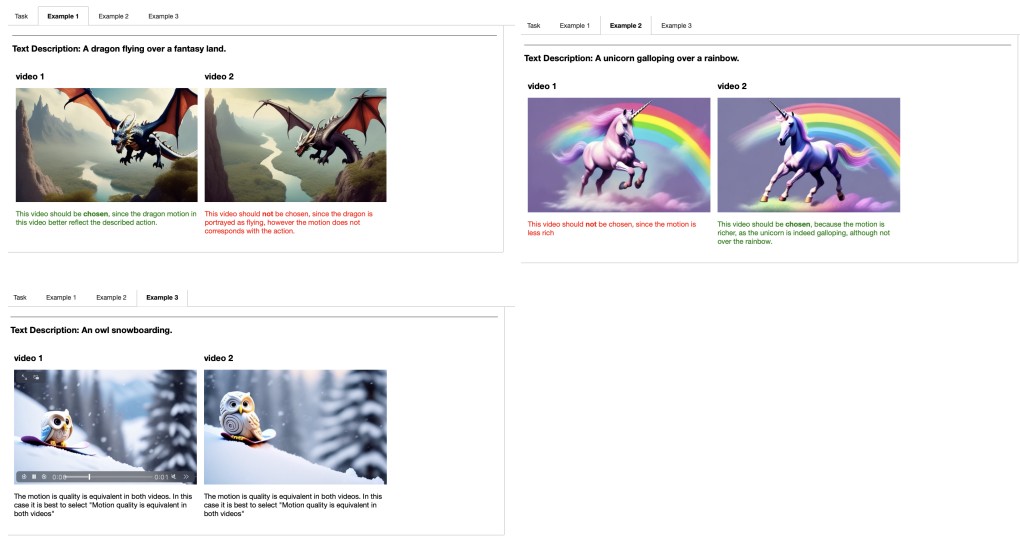

Figure 16: Examples provided in the user study for text-motion alignment.

