# OpenReview forum: "Multi-Shot Character Consistency for Text-to-Video Generation"
_ICLR.cc/2025/Conference — Submitted to ICLR 2025_

### Official Review · Reviewer_ntRv · 2024-10-30

**Soundness:** 2
**Presentation:** 1
**Contribution:** 2
**Rating:** 5
**Confidence:** 4

**Summary:**

This paper aims to generate multi-shot videos with consistent characters in a zero-shot manner. They claim that there is a trade-off between preserving character identity and video dynamics, thereby designing a two-phase approach, Q-preservation and Q-Flow, to balance the two respects.

**Strengths:**

1. **Training-Free Approach for Subject Consistency Across Shots**: This work offers a training-free method for generating subjects with consistent identity across varying scenarios and shot transitions, which is valuable for practical applications where maintaining coherence in subject appearance is essential.

2. **Novel Insights on Self-Attention Query Features**: The authors provide fresh insights into the role of self-attention query features, demonstrating that these features effectively capture both motion and identity.

3. **Query-Preservation and Q-flow Techniques**: By preserving query features during early denoising and applying a tokenflow-inspired approach to select keyframes, the method achieves partial injection of query features to adjacent frames. Although it draws heavily from ConsisStory and TokenFlow, this approach has demonstrated effectiveness in enhancing subject consistency and motion dynamics to a certain extent.

**Weaknesses:**

1. **Limited Novelty in Video Storyboarding**: The innovation of the proposed video storyboarding approach is limited. The primary method relies on frame-wise SDSA, which largely mirrors the approach used in ConsiStory. The only notable difference lies in the mask source, utilizing CLIPseg and OTSU segmentation rather than cross-attention.

2. **Poor Writing and Project Organization**: The paper's writing and the project page's layout hinder comprehension, making it difficult for readers to follow the key contributions the authors intend to convey.

3. **Minimal Improvement over Baseline Models**: The generated video storyboarding results appear similar to those produced by existing video generation baselines like Videocrafter2 or TokenFlow encoder, with little noticeable difference in output quality.

4. **Lack of Motion Dynamics**: The method demonstrates limited motion dynamics. In most video segments, the objects remain static, and in every case, the object consistently occupies the center of the frame, resulting in rigid, uninspired visuals.

5. **Overclaiming the Benchmark**: The authors’ claim of establishing a benchmark based on a dataset of only 30 videos, each containing 5 video shots, is unsubstantiated. This dataset is insufficiently sized and lacks diversity, with evaluations limited to character consistency and dynamic degree, providing a narrow view that does not comprehensively assess the model's capabilities.

**Questions:**

1. **Inconsistencies in Object Appearance and Action**: In the ablation study on query preservation (`ablation_q`), inconsistencies persist. For example, in the first video, the right hand of the Muppet character appears red, while in the third shot, it is not. Additionally, although the Muppet is intended to perform aerobics in a Sesame Street setting, it merely flicks its hand briefly, failing to convey the intended action sequence.

2. **Static Object Issues in ConsiStory Component Ablation Study**: In the ablation study on ConsiStory components for video generation, the rabbit character intended to surf, train, and ride a bike appears mostly static in the first and third shots. This raises the question of whether these issues stem from limitations in the base model’s dynamic capabilities. If so, would using models with stronger dynamic performance, such as Dynamic Crafter or CogVideo, potentially improve motion consistency and address these static object limitations?

If the video dynamic problem is addressed, I am willing to increase my score.

**Details Of Ethics Concerns:**

No Ethics Concerns

---

> ### Author Response · Authors · 2024-11-24
> **Thank you for the review!**
>
> Thank you for your detailed feedback and thoughtful suggestions, as well as your willingness to increase your score based on improvements. Your comments on motion dynamics, novelty, and clarity have been instrumental in refining our work. In particular, your suggestion to evaluate our method using a stronger video model inspired us to conduct new experiments with T2V-Turbo-V2, yielding significantly enhanced motion dynamics. Below, we address your comments in detail.
>
> * **Lack of Motion Dynamics ..  do these issues stem from limitations in the base model’s dynamic capabilities. If so, would using models with stronger dynamic performance … If the video dynamic problem is addressed, I am willing to increase my score.**
>
>     Following the reviewer's request, we applied our method to T2V-Turbo-V2 [Li et al., NeurIPS 2024], a recent state-of-the-art video model released in October 2024 that offers significantly larger motion dynamics and improved video quality. Results from this experiment—both qualitative (Figure 8, Appendix) and quantitative (Table 1, Appendix)—show that combining our approach with Turbo-V2 produces consistent subjects while achieving much more dynamic visuals. Specifically, the dynamic degree **improves threefold**, from 20 to 62, while maintaining the same level of text alignment. This improvement is also evident in the qualitative examples the reviewer ask about, featuring the rabbit and Muppet characters (see Figure 8, Appendix and the online website). There, our method combined with Turbo-V2 demonstrates noticeably enhanced motion dynamics compared to its application with VideoCrafter2.
>
>     The static motion in Ours+VideoCrafter2 results reflects the limitations of the underlying pretrained model. However, our method still achieves a better consistency-dynamics balance than naive consistency approaches, as shown in the ablation studies (Figures 4 and 5) and the user study.
>
>     [Li et al.], T2V-Turbo: Breaking the Quality Bottleneck of Video Consistency Model with Mixed Reward Feedback, NeurIPS 2024
>
>
> * **Novelty.**
>
>     Our key novelties are twofold: (1) we demonstrate that query features (Q) encode both motion and identity—*this is our main insight*. This insight directly motivates (2) our two-phase query injection mechanism, which is specifically designed to address the challenges arising from this observation, particularly in preserving identity without compromising motion. For that, we adopt components from prior work in non-trivial ways to form a cohesive solution.
>
>     Furthermore, our work is the first to address this problem setup for video generation and the unique challenges it presents. Indeed, we also introduce additional technical improvements, such as the frame-wise SDSA and the use of estimated clean image (pred-x0) for mask generation, rather than relying on internal network activations used in prior methods.
>
> * **Clarity.**
>
>     We thank the reviewer for this valuable feedback. We have improved the paper's clarity through:
>
>     - **Added Notations Section**: A dedicated Notations section defining key terms (e.g., $Q_v$, $Q_c$, $Q_f$) to reduce ambiguity.
>
>     - **Aligned Structure**: Revised the methods section to mirror ConsiStory's three steps using  clear bold headers, making connections to prior work explicit: (1) SDSA, (2) Layout Diversity, (3) Refinement Injection.
>
>     - **Focused Content**: Technical derivations and complex details were moved to the appendix, while the main text emphasizes core challenges before presenting the solutions.
>
>     The project page has also been updated for better navigation.
>
> * **Improvement over Baseline Models.**
>
>     While we acknowledge that the method is not perfect and there is room for improvement in future work, we emphasize that it already achieves significant advancements over existing baselines. For instance, its consistency is rated higher by 66% of users compared to the best baseline and by 79% of users compared to the base model, all while maintaining competitive text-motion alignment.
>
> * **Overclaiming the Benchmark.**
>
>
>     We revised and downplayed the benchmark claim to reflect the scope of our evaluation. Our evaluation, based on 150 generated videos (see L431, L471), provides distinguishable error bars and ensures robust evaluation results. In response to this comment, we also report additional semantic alignment scores, including “text-similarity” (referred to as “overall-consistency” by VBench) and VBench’s “subject-consistency,” which measures the similarity between frames within the same video shot using DINO (see Table 1 in the Appendix).

---

> > ### Comment · Reviewer_ntRv · 2024-11-27
> >
> > Thank you for your response and additional experiments. While T2V-Turbo-V2 improves motion dynamics, I noticed issues like background removal (e.g., missing streets during skating in Sesame Street and train tracks while surfing on a train in Rabbit Toy). The characters and story still lack realism, and challenges in multi-shot character-consistent story generation remain, such as duration, natural motion, and story richness. I will maintain my original score.

---

> > > ### Author Response · Authors · 2024-11-28
> > >
> > > Thank you for your feedback. We appreciate your thorough evaluation of our work and the opportunity to clarify our objectives and results.
> > >
> > > We wish to emphasize that the issues you mentioned—missing backgrounds (streets/tracks), lack of realism, unnatural motion, and the limited duration of shots—are inherent limitations of the base T2V-Turbo-V2 model, which our study does not aim to address. Our primary objective is to enhance subject consistency across different shots, rather than directly improving visual quality or realism.
> > >
> > > We demonstrated that our approach can provide more significant improvements when applied to a stronger baseline model that inherently addresses one of these concerns. The additional experiment with T2V-Turbo-V2, as you requested, shows that when a more advanced base model is available, our method indeed enhances subject consistency in conjunction with the improvements stemming from that model (motion dynamics in this case).
> > >
> > > Moreover, we have introduced a new section on our project website titled “Background details in dynamic video-models”, where you can observe that the issues you mentioned stem from the vanilla T2V-Turbo-V2, rather than from our method. In these results, you can view our generation results alongside those from the base T2V-Turbo-V2 model. In these comparisons, we use hyperparameters specifically tuned to preserve detail, which enables our approach to better maintain the background and detail quality of the base model while maintaining subject consistency.
> > >
> > > Thank you for considering these points. We hope this clarifies our contributions and the potential of our method in conjunction with future advancements in base models.
> > >
> > > * The project page was updated before the deadline for updating the paper / results

---

### Official Review · Reviewer_yj9C · 2024-10-30

**Soundness:** 3
**Presentation:** 4
**Contribution:** 3
**Rating:** 8
**Confidence:** 4

**Summary:**

This paper proposes a method called "Video Storyboarding" to generate multi-shot videos with consistent characters across different scenes while preserving motion quality.

**Strengths:**

1. Framewise Subject-Driven Self-Attention to maintain consistency without compromising motion
2. Novel two-phase query injection strategy to balance identity preservation and motion quality
3. Adaptation of refinement feature injection to both conditional and unconditional denoising steps

**Weaknesses:**

1. Approach limited to short video clips, unsure how it would scale to longer videos
2. Balancing identity preservation and motion quality is still challenging, with potential tradeoffs

**Questions:**

Overall, this is a well-designed and rigorously evaluated method that represents a significant advancement in generating coherent multi-shot video sequences. The authors have done a commendable job in addressing the complex challenge of maintaining character consistency while preserving natural motion.

Some questions for the authors:

1. Have you explored any strategies to further improve the balance between identity preservation and motion quality? Are there other techniques beyond query injection that could be investigated?
2. How do you envision this approach scaling to longer video sequences? What additional challenges might arise, and how could the method be adapted to handle them?
3. The user study results showed that the ConsiS Im2Vid baseline achieved the highest set consistency among the baselines. Can you comment on the strengths of this approach and how it might be combined or compared with your Video Storyboarding method?

---

> ### Author Response · Authors · 2024-11-24
> **Thank you for the review and support!**
>
> Thank you for your thoughtful feedback, insightful questions, and support of our work. We greatly value your perspectives on scaling to longer videos, balancing identity preservation and motion quality, and combining methods like ConsiS Im2Vid with our approach. Below, we provide detailed responses to your questions.
>
>
> * **How do you envision this approach scaling to longer video sequences? What additional challenges might arise, and how could the method be adapted to handle them?**
>     In long video sequences, subjects undergo viewpoint variations, causing their appearance to evolve due to perspective changes. Extended attention mechanisms, while useful for maintaining consistency, often promote a single dominant viewpoint. This suppresses viewpoint diversity and hinders the ability to handle natural perspective shifts over time.
>
>     One way to address this is to apply extended attention selectively, instead of connecting frames in a non-specific manner. For this, the video can be divided into shorter temporal segments, and attention is extended only across segments detected to share similar viewpoints. This approach may maintain subject consistency within segments while respecting viewpoint diversity across the sequence.
>
> * **Have you explored any strategies to further improve the balance between identity preservation and motion quality? Are there other techniques beyond query injection that could be investigated?**
>     Yes, there is definitely room for improvement in balancing identity preservation and motion quality. One approach we explored, though it did not yield significant improvements, was a training-free rank-one update to the network weights by copying average features from anchor videos, inspired by [Bau et al. ECCV 2020].
>
>     Several potential strategies for improvement include leveraging temporal attention layers to link features across video shots, such as copying temporal features from one shot to another. Alternatively, generating a single long video, as done in VSTAR [Li et al. 2024], could help maintain stronger identity preservation. However, the challenge lies in creating clear-cut transitions between scenes, avoiding the scene-transition style of VSTAR videos, while ensuring alignment with text prompts.
>
>     Another avenue is a training-based approach, such as those explored in works like [Zeng et. al CVPR 2024], which could allow for deeper optimization of the balance between identity and motion dynamics.
>
>     References:
>     [Bau et. al] Rewriting a Deep Generative Model, ECCV 2020
>
>     [Li et. al] Generative Temporal Nursing for Longer Dynamic Video Synthesis, preprint 2024
>
>     [Zeng et. al] JeDi: Joint-Image Diffusion Models for Finetuning-Free Personalized Text-to-Image Generation, CVPR 2024
>
> * **Comment on the strengths of ConsiS Im2Vid baseline approach and how it might be combined or compared with your Video Storyboarding method?**
>     The strength of the ConsiS Im2Vid baseline lies in its ability to generate video shots from a set of consistent reference images. However, it has key limitations: (1) it lacks a mechanism to enforce cross-shot consistency, leading to identity variations, even when conditioned on images with consistent subjects, and (2) it lacks text-prompt control, resulting in static or misaligned motion.
>
>     Combining Im2Vid with our Video Storyboarding method is an excellent idea that could lead to a robust hybrid approach. Framewise-SDSA and feature refinement could enhance subject consistency across shots. That said, the challenge of injecting motion remains, as Im2Vid lacks text-prompt control. This could be addressed by generating a non-consistent video using a text-to-video model and applying an inversion technique to extract and inject motion activations into the Im2Vid framework. Alternatively, combining our approach with a pretrained model conditioned on both image and text (image+text-to-video) could enable simultaneous control over subject identity and motion dynamics.

---

### Official Review · Reviewer_kLfg · 2024-11-03

**Soundness:** 3
**Presentation:** 2
**Contribution:** 2
**Rating:** 5
**Confidence:** 3

**Summary:**

This paper presents Video Storyboarding, a training-free method that enhances pre-trained text-to-video models to generate multiple shots with consistent characters while maintaining high video quality and responsiveness to text prompts. By leveraging self-attention query features that capture motion and identity, the method addresses the trade-off between character consistency and video dynamics through a novel query injection strategy. Experimental results show significant improvements in character consistency and motion quality, offering insights into the video generation process and the interplay of structure and motion in diffusion models.

**Strengths:**

1. This paper introduces a training-free method to ensure charactoer consistency and motion adherence in producing multi-shot video sequences.
2. This paper presents a two-phase query injection strategy to balance encoding motion and identity.
3. A benchmark and evalution protocol are proposed to evaluate consistency of video generation.

**Weaknesses:**

1. The conducted experiments are not comprehensive, including two aspects: (a) The paper only provides several comparison samples. (b) The paper misses some important baseline methods, e.g., VSTAR[1]
2. Although the purpose of the paper is to maintain consistency of characters across different video clips, the results are not particularly good. For example, in Fig.3, the color and style of clothes change across different video shots.

[1] Li, Yumeng, William Beluch, Margret Keuper, Dan Zhang, and Anna Khoreva. "VSTAR: Generative Temporal Nursing for Longer Dynamic Video Synthesis." arXiv preprint arXiv:2403.13501 (2024).

**Questions:**

1. The paper only demonstrates the performance of a single character across different videos, and the reviewer is curious about how the proposed  method performs with multiple characters.
2. The prompt in the paper provides overly detailed descriptions of the character. Would a more concise description impact character consistency? For example, replace the "Cinematic, middle-aged female athlete" in Fig.8 with "A woman".

---

> ### Author Response · Authors · 2024-11-24
> **Thank you for the review!**
>
> Thank you for your constructive feedback and thoughtful suggestions, which have helped us improve the scope of our work. We appreciate your detailed comments on expanding baseline comparisons, exploring concise prompts, and evaluating multiple characters, as they helped us strengthen both our experiments and presentation. In particular, your points regarding VSTAR and broader evaluations led us to incorporate new results, which we believe address your concerns. Below, we respond to your comments in detail.
>
> * **Experiments: The paper misses some important baseline methods, e.g., VSTAR[1]. The paper only provides several comparison samples.**
>
>
>     We first note that VSTAR's code was released on October 10, after the ICLR submission deadline. However, in response to the reviewer’s request, we have now included both qualitative and quantitative comparisons to the VSTAR baseline. Please see the results in the Appendix (Table 1, Figure 11 and the online website). Overall, while VSTAR produces large motion dynamics, it struggles with prompt-specific control, often resulting in entire videos misaligning with text descriptions. Since it achieves consistency through continuous video generation, VSTAR is better suited for scene transitions rather than independent video shots.
>
>     Additionally, regarding the scope of comparisons, our user study and automated metrics were conducted with 150 generated videos (see L431, L471). This sample size provides distinguishable error bars, ensuring robust evaluation results.
>
> * **Would a more concise description impact character consistency? For example … "A woman".**
>
>     Following the reviewer’s request, we have added qualitative results demonstrating our method’s ability to handle general subject categories (see Appendix Figure 9 and the online website). These examples show that our approach can successfully generate consistent videos for broad subject types like "woman" and "rabbit," indicating strong performance even with concise, superclass-level prompts.
>
>     Overall, a more concise description generally increases the gap between our method and the baselines, as it amplifies variation between shots in the baseline methods. Note that we intentionally adopted prompts with a complexity level comparable to consistent-character image generation methods, such as Consistory. For example, prompts like "Cinematic, middle-aged female athlete" are not overly complex, especially when compared to the highly detailed descriptions used in works like Sora (e.g., “A stylish woman walks down a Tokyo street... She wears a black leather jacket, a long red dress, and black boots, and carries a black purse. She wears sunglasses and red lipstick.” or “the 30-year-old space man wearing a red wool knitted motorcycle helmet”).
>
> * **How the proposed method performs with multiple characters.**
>
>     Following the reviewer’s request, we have added qualitative results demonstrating the capability to handle multiple subjects (see Appendix Figure 10 and the online website). By incorporating two subjects in the prompt of the zero-shot mask, our approach can consistently render multiple characters within the same scene, as illustrated by examples featuring girl-owl and boy-teddy bear pairs.
>
> * **Quality of consistency results .. in Fig.3, the color and style of clothes change across different video shots.**
>
>     We acknowledge that the method is not perfect and that there is room for improvement in future work. However, it is worth noting that in some cases, changes in color and style, such as the girl wearing different attire while surfing, align with the context of the action and user expectations. A rigid preservation of clothing or style across all shots might not always be desirable.
>
>     That said, our method already provides significant improvements over existing baselines. For instance, its consistency is rated higher by 66% of users compared to the best baseline and by 79% of users compared to the base model, all while maintaining competitive text-motion alignment.

---

> > ### Author Response · Authors · 2024-11-30
> > **Follow up**
> >
> > Dear Reviewer kLfg,
> >
> > We sincerely appreciate the time and effort you have dedicated to reviewing our work.
> >
> > We hope you had an opportunity to review our response from November 24. In it, we included comparisons with VSTAR, as you suggested, demonstrated results with concise prompts and multiple characters, and provided additional context for our consistency results.
> >
> > Are there any other concerns or questions we can help address? We would be happy to provide further clarification.
> >
> > Thank you,
> >
> > The authors

---

> > ### Comment · Reviewer_kLfg · 2024-12-01
> >
> > Thank you for the detailed reply. I have carefully read your response and the experimental results, and I have decided to maintain my original score. The main reasons are that the method's performance is not satisfactory: (1) Consistency is still relatively poor on simple objects, such as the woman in Figure 9. (2) The generated video movements are either too subtle or do not match the text description, such as the "Feeding" in Figure 10.

---

> > > ### Author Response · Authors · 2024-12-02
> > >
> > > Thank you for taking the time and effort to review our work and provide thoughtful feedback. We value your insights and have carefully considered your comments.
> > >
> > > We acknowledge the concern about motion subtlety. Our initial model was indeed limited in this aspect due to its reliance on VideoCrafter2 (e.g., see VideoCrafter2 in Figure 2 - "playing Wii" / "playing w. trees", Figure 11 - athlete). Following another reviewer's suggestion, we experimented with T2V-Turbo-V2, achieving more dynamic visuals while maintaining consistency (Table 1-appendix, Figure 8). These new results are also shown on our website under "Video Storyboarding with a stronger pretrained model, T2V-Turbo-V2".
> > >
> > > In the broader context, our experiments with T2V-Turbo-V2 demonstrate how our approach scales to stronger base models. This enables motion preservation during consistent multi-shot video generation, which is particularly relevant as text-to-video synthesis continues to evolve.
> > >
> > > Regarding consistency, we acknowledge that the method is not perfect. However, for the first time, it enables consistent multi-shot video generation with measurable, notable improvements over baseline methods, both qualitatively and quantitatively. Notably, the baseline methods exhibit very limited performance, which renders them largely ineffective in this task. Empirical evidence from our user study reveals that participants preferred our outputs twice as often as those from the best baseline model (66%) and four times as often as those from the base model (79%).
> > >
> > > Thank you for considering these points. We hope this clarifies our contributions and the potential of our method in conjunction with future advancements in base models.

---

### Official Review · Reviewer_e8uT · 2024-11-03

**Soundness:** 3
**Presentation:** 2
**Contribution:** 2
**Rating:** 5
**Confidence:** 4

**Summary:**

This paper targets the problem of character consistency in text-to-video generation. The authors propose a training-free method to solve this problem. They find the query in the attention encodes the information of both motion and identify, which leads to the trade-off between motion dynamics and identity consistency. The experimental model used is VideoCrafter2. To solve the trade-off problem, they propose a new query injection method. Specifically, they share features between different video clips. Then, they replace the Q (query) with those from the original generation (to maintain motion from the original generation). After that, they leverage the flow map from vanilla keyframes to guide the Q injection. Their results achieve the character consistency while keeping the original motion dynamics and text alignment. The text alignment is evaluated via user study. The overall metrics for evaluation are three aspects: motion degree, id consistency, and motion text alignment.

**Strengths:**

1. The method is tuning-free and does not require any further training.
2. The results outperform baseline methods. Some of the provided visual results look good.

**Weaknesses:**

1. The paper is relatively hard to follow regarding the details of the method part.
2. Lack of novelty: 1. The id presevering mechanism is built upon the SDSA. the SDSA is adopted from ConsiStory [Tewel et al. 2024] with two minor modifications: (1) The attention not attend to each other with all frames from different clips, but one single frame from each clip. (2) The mask estimation use ClipSeg, rather than estimated from the cross attention. 2. The motion preserving is leveraging TokenFlow [Geyer et al. 2023] to inject the motion based on the flow from original keyframes. Thus, the method is like a A+B combination with some minor modifications.
3. The key insight "self-attention query features (O) encode both motion and identity" lack experimental results to demonstrate.
4. The results are not perfect, e.g., inconsistent hairstyles in the 3rd row of Figure 1.
5. The evaluation does not contain the overall video generation quality and the qualitative semantic alignment scores.
6. Minor formate issues like inconsistent figure reference:  Figure 1 and Fig. 4; And strange line break at line 244 and line 291.

**Questions:**

1. Does the overall generation quality decrease after the proposed method?
2. How does the motion quality changed after the proposed method?

---

> ### Author Response · Authors · 2024-11-24
> **Thank you for the review!**
>
> Thank you for your valuable feedback and thoughtful insights. They have  greatly helped us refine the clarity, novelty, and evaluation of our work. We appreciate your recognition of our key insight that “query features encode both motion and identity.” We hope that our revisions, including clearer explanations of our method, additional experiments, and expanded evaluations, address your concerns effectively. Below, we address your concerns in detail. Below, we respond to your points in detail.
>
>
> * **Novelty**
>
>     Our key novelties are twofold: (1) as the reviewer highlighted in his point #3, the insight that "query features (Q) encode both motion and identity." This key observation directly inspired (2) our two-phase query injection mechanism specifically designed to address the challenges arising from this observation, particularly in preserving identity without compromising motion. We respectfully disagree with the notion that our method is merely an "A+B combination with minor modifications.". While the method builds on components from prior work, they are adapted in non-trivial ways to form a cohesive solution.
>
>     Furthermore, our work is the first to address this problem setup for video generation and the novel challenges it presents. Indeed, we also introduce additional technical improvements, such as the frame-wise SDSA and the use of estimated clean image (pred-x0) for mask generation, rather than relying on internal network activations used in prior methods.
>
> * **Clarity of the method part.**
>
>     We thank the reviewer for this valuable feedback. We have improved the paper's clarity through:
>
>     - **Added Notations Section**: A dedicated Notations section defining key terms (e.g., $Q_v$, $Q_c$, $Q_f$) to reduce ambiguity.
>
>     - **Aligned Structure**: Revised the methods section to mirror ConsiStory's three steps using  clear bold headers, making connections to prior work explicit: (1) SDSA, (2) Layout Diversity, (3) Refinement Injection.
>
>     - **Focused Content**: Technical derivations and complex details were moved to the appendix, while the main text emphasizes core challenges before presenting the solutions.
>
> * **The key insight "self-attention query features (Q) encode both motion and identity" lack experimental results to demonstrate.**
>
>     Please see the experiment in Figure 4 and the discussion around L375. We demonstrate that injecting the query features from the vanilla videos results in strong motion preservation. At the same time, it causes a *loss of subject identity* (e.g., the Muppet’s color change), clearly indicating that the Q tokens encode both motion and identity information.
>
> * **The evaluation does not contain the overall video generation quality and the qualitative semantic alignment scores.**
>
>     In response to this comment, we now provide additional semantic alignment scores, including “text-similarity” (referred to as “overall-consistency” by VBench) and VBench's “subject-consistency,” which measures the similarity between frames within the same video shot using DINO (see Table 1 in the Appendix). We also note that our video generation quality scores are similar to those of the underlying pretrained model (see discussion in L474), which is why we focused on presenting two key metrics — multi-shot set consistency and dynamic degree — that are more distinguishable relative to the pretrained model.
>
> * **Results are not perfect, e.g., inconsistent hairstyles.**
>
>     We acknowledge that the method is not perfect and there is room for improvement in future work. However, we highlight that our method already provides a significant improvement when compared to existing baselines. For instance, its consistency is rated higher by 66% of users compared to the best baseline and by 79% of users compared to the base model, all while maintaining competitive text-motion alignment.
>
> * **Does the overall generation quality decrease after the proposed method?**
>
>     No, the overall generation quality does not decrease. Only the generated motion is somewhat reduced (e.g. the dynamic degree) when using the VideoCrafter2 pretrained model (see Figure 6). Following Reviewer ntRv's comment, we included additional results using another SoTA pretrained video model (T2V-Turbo-V2). In this case, the dynamic degree is only slightly affected (see Table 1 in the appendix).
>
> * **How does the motion quality changed after the proposed method?**
>
>     According to the user study (Figure 7, right), 55% of our generated motions were judged to be of similar or better quality compared to the vanilla model.

---

> > ### Author Response · Authors · 2024-11-30
> > **Follow up**
> >
> > Dear Reviewer e8uT,
> >
> > We sincerely appreciate the time and effort you have dedicated to reviewing our work.
> >
> > We hope you had an opportunity to review our response from November 24. In it, we provided additional evaluation metrics, described how we improved the paper's clarity, clarified how Figure 4 and L375 support our key insight about query features, and elaborated on our paper's novelty. We also addressed your questions about generation quality.
> >
> > Are there any other concerns or questions we can help address? We would be happy to provide further clarification.
> >
> > Thank you,
> >
> > The authors

---

### Author Response · Authors · 2024-11-24
**General response to reviewers**

We thank all the reviewers for their useful and insightful feedback. We are encouraged that the reviewers found our work to offer “**novel insights**” into structure and motion (ntRv, kLfg) and commended our “**novel two-phase query injection strategy**” (yj9C, kLfg, ntRv). The reviewers appreciated our “**valuable**,” tuning-free problem setup (**all**) and found our work to be “**sound**” (e8uT, kLfg, yj9C), while acknowledging our method as “**effective**” (ntRv), with “**significant improvements**” (kLfg), “**outperform baseline**” (e8uT) and being “**rigorously evaluated**” (yj9C).


Our work has benefited tremendously from your feedback. Below are the main modifications to the manuscript (colored blue in the updated PDF):

1. **Stronger pretrained model, T2V-Turbo-V2**

    Following Reviewer ntRv’s suggestion, we tested our method with T2V-Turbo-V2 [Li et al. NeurIPS 2024], a state-of-the-art video model with enhanced motion capabilities. Results (Appendix: Table 1, Figure 8) show a threefold improvement in dynamic degree (from 20 to 62) while maintaining text alignment and subject consistency.

2. **Expanded Experimental Results and Qualitative Demonstrations**

    - **VSTAR Baseline Comparisons**: We added qualitative and quantitative results comparing our method with VSTAR (Appendix: Table 1, Figure 11), as suggested by Reviewer kLfg.
    - **Semantic Alignment Metrics**: New results include text-similarity and subject-consistency scores (Appendix: Table 1), as suggested by Reviewers (e8uT, ntRv).
    - **Additional Qualitative Results**: We demonstrated our method’s ability to handle general subject categories like “woman” or “rabbit” (Appendix: Figure 9) and multiple subjects (Appendix: Figure 10), as suggested by Reviewer kLfg.

3. **Improved Clarity**:

    We refined the paper’s writing, as suggested by Reviewers (e8uT, ntRv). Specifically, we added a dedicated *Notations Section* to define key terms, and revised the methods section to align with ConsiStory's three-step structure. Technical derivations and complex details were moved to the appendix to ensure the main text focuses on core challenges and solutions.


We look forward to addressing any remaining questions from the reviewers and continue engaging in further discussion. If you find our response satisfactory, we kindly ask you to consider raising your rating in recognition of our core contributions.

---

### Author Response · Authors · 2024-12-04
**A final note**

Dear Reviewers and AC,

We sincerely thank you for your valuable time and insightful feedback, which has greatly benefited our work.

In a final note, we wish to emphasize that 3 of 4 reviewers found our approach to be “**sound**”, offering “**novel insights**” into the representation of motion, structure and identity (ntRv, kLfg) and commended our “**novel two-phase query injection strategy**” (yj9C, kLfg, ntRv).

In a broader context, for the first time, our work enables consistent multi-shot video generation, combined with motion-preservation. Notably, the baseline alternatives exhibit very limited performance, both qualitatively and quantitatively, which renders them largely ineffective in this task. Empirically, users had 2-4x preference over the alternatives (66-79%). Additionally, our new experiments with T2V-Turbo-V2 demonstrated that limitations such as video quality and motion issues diminish as our approach scales to stronger models.

We are confident that our work contributes to the dialogue in the field, advancing the understanding of motion, structure and identity in diffusion models,.

Thank you for considering our work.

Best regards,

The Authors

---

### Meta-Review · Area_Chair_QrUP · 2024-12-16

**Metareview:**

This paper presents a training-free approach to ensure character consistency and motion adherence in generating multi-shot video sequences.

The paper received mixed ratings (5, 5, 5, 8) after rebuttal, with key concerns revolving around clarity of writing (e8uT, ntRv), novelty of the method (e8uT, ntRv), and quality of results (e8uT, kLfg, ntRv).

In the post-rebuttal discussion, reviewers acknowledged that the revised version improved clarity. However, concerns about the novelty and quality of results remained. Specifically:
- The motivation of “query features encode both motion and identity” was interesting, but was not convincingly demonstrated through the results.
- The proposed method offered only incremental innovations compared to ConsiStory and TokenFlow.
- Despite including T2V-Turbo, the results failed to address issues with motion consistency, video fidelity, and multi-shot consistency.

The AC agrees with the majority of the reviewers and recommends rejecting the paper.

**Additional Comments On Reviewer Discussion:**

The paper received mixed ratings (5, 5, 5, 8) after rebuttal, with key concerns revolving around clarity of writing (e8uT, ntRv), novelty of the method (e8uT, ntRv), and quality of results (e8uT, kLfg, ntRv).

In the post-rebuttal discussion, reviewers acknowledged that the revised version improved clarity. However, concerns about the novelty and quality of results remained. Specifically:
- The motivation of “query features encode both motion and identity” was interesting, but was not convincingly demonstrated through the results.
- The proposed method offered only incremental innovations compared to ConsiStory and TokenFlow.
- Despite including T2V-Turbo, the results failed to address issues with motion consistency, video fidelity, and multi-shot consistency.

The reviewer yj9C with a positive score submitted short comments and did not participate in any discussion.

---

### Decision · Program_Chairs · 2025-01-22

Reject